# An Emerging Disease of Chickpea, Basal Stem Rot Caused by *Diaporthe aspalathi* in China

**DOI:** 10.3390/plants13141950

**Published:** 2024-07-16

**Authors:** Danhua Wang, Dong Deng, Junliang Zhan, Wenqi Wu, Canxing Duan, Suli Sun, Zhendong Zhu

**Affiliations:** Key Laboratory of Grain Crop Genetic Resources Evaluation and Utilization, Institute of Crop Sciences, Chinese Academy of Agricultural Sciences, Beijing 100081, China; wdh1925798019@163.com (D.W.); 82101219111@caas.cn (D.D.); zhanjunliang1170@163.com (J.Z.); wuwenqi@caas.cn (W.W.); duancanxing@caas.cn (C.D.)

**Keywords:** Chickpea, *Diaporthe aspalathi*, emerging disease, *Diaporthe*/*Phomopsis* spp.

## Abstract

Chickpea (*Cicer arietinum* L.) is an important legume crop worldwide. An emerging disease, basal stem rot with obvious wilt symptoms, was observed in the upper part of chickpea plants during the disease survey in Qiubei County of Yunnan Province. Three fungal isolates (ZD36-1, ZD36-2, and ZD36-3) were obtained from the diseased tissue of chickpea plants collected from the field. Those isolates were morphologically found to be similar to *Diaporthe aspalathi*. Molecular sequence analyses of multiple gene regions (ITS, *tef1*, *tub2*, *cal*, and *his3*) indicated that the three isolates showed a high identity with *D. aspalathi*. Pathogenicity and host range tests of the isolates were performed on the original host chickpea and eight other legume crops. The isolates were strongly pathogenic to chickpea and appeared highly pathogenic to soybean, cowpea, and mung bean; moderated or mild pathogenic to adzuki bean and common bean; however, the isolates did not cause symptoms on grass pea (*Lathyrus sativus*). *Diaporthe aspalathi* was previously reported as a main pathogen causing the southern stem canker in soybean. To our knowledge, this is the first report of *D. aspalathi* inducing basal stem rot on chickpea worldwide.

## 1. Introduction

Chickpea (*Cicer arietinum* L.) is a crucial legume crop cultivated for its nutritional significance, adaptability, and economic value [1]. Chickpea ranked third with an annual production of approximately 11.6 million tons, following beans and peas in worldwide pulse production in 2020 [2]. With a rich history of consumption dating back thousands of years, chickpea holds a prominent place in various cuisines, particularly in regions across the Mediterranean, Middle East, and South Asia. Beyond its culinary appeal, chickpea plays an important role in sustainable agriculture as a nitrogen-fixing crop, contributing to soil fertility and crop rotation practices [3].

Chickpea is affected by a range of pathogens, including fungi, bacteria, viruses, mycoplasma, nematodes, and Viroid [4,5]. Among these pathogens, fungal species are predominantly responsible for major chickpea diseases. Major fungal diseases causing substantial yield losses worldwide include Ascochyta blight, Fusarium wilt, powdery mildew dry root rot, Botrytis gray mold, and wet root rot. Therefore, controlling fungal disease is crucial to chickpea production [6].

*Diaporthe/Phomopsis* spp. are known to be associated with a wide range of host plants, about 167 plant species, as pathogens, endophytes, or saprobes [7,8,9]. The *Diaporthe/Phomopsis* complex (DPC) has been recognized as one of the major limiting factors on many crops worldwide due to inducing several devastating diseases [9,10]. The in-depth studies of *Diaporthe/Phomopsis* spp. were carried out in several host crops, especially those associated with soybean [10,11], citrus [12,13,14], sunflower [7,15], kiwifruit [16], and grape [17]. *Diaporthe/Phomopsis* spp. was first reported to cause pod and stem blight diseases on the most important legume crop, soybean, in the USA [18]. Since then, they occurred in other regions of the USA and almost all soybean-producing countries worldwide [15], Brazil [19], Argentina [20], Croatia [21], Korea [11], Canada [22], and China [23,24,25].

The *Diaporthe*/*Phomopsis* complex (DPC) induced severe diseases in soybean, including southern and northern stem canker, pod and stem blight, stem blight, and seed decay [11,22,23,24,26]. The typical symptoms of stem canker included foliar interveinal chlorosis and necrosis, reddish-brown and sunken lesions stems, protruding beaks perithecia [21,27]. The most obvious sign of pod and stem blight is the presence of small and black pycnidia in linear rows on infected material [25]. The typical characteristic symptoms of stem blight were dry rot with discoloration on the lower stem at the junction of a branch or petiole, which caused plant wilt and death finally [23]. The typical characteristic of Phomopsis seed decay is cracked and shriveled with white mold on the surface of the seeds; sometimes, infected seeds do not show any symptoms [11,22]. *P. longicolla* (*Diaporthe longicolla*) is the primary cause of Phomopsis seed decay [28,29]. Although Phomopsis seed decay is caused primarily by *D. longicolla*, other Diaporthe species were also reported to be associated with soybean seed decay, such as *D. sojae* (syn. *D. phaseolorum* var. *sojae*), *D. caulivora*, *D. aspalathi*, *D. eres*, *D. novem* (syn. *D. pseudolongicolla*), *D. foeniculina*, and *D. rudis* [30]. *D. sojae* is the primary causal agent of pod and stem blight [29]. *P. longicolla* can also induce pod and stem blight. The two forms of soybean stem canker, north stem canker and south stem canker, are caused by *D. caulivora* (syn. *Diaporthe phaseolorum* var. *caulivora*) and *D. aspalathi* (syn. *Diaporthe phaseolorum* var. *meridionalis*), respectively [11,19,27]. *D. caulivora* can also induce Phomopsis seed decay on soybean [11]. *D. novem* and *D. eres* showed strong virulence to soybean stems while showed moderate and no virulence to seeds, respectively [10]. Other DPC species, such as *D. citri*, *D. melonis*, and *D. endophytica,* have also been recorded to be obtained from soybean stems or seeds, but their virulence to soybean was still uncertain [21].

Identification of DPC species in history has been based on morphological differentiations, including the presence of an anamorph or a teleomorph, colonial and conidia morphology, disease symptoms, and virulence on soybean [31,32]. However, because of the diversity of DPC morphological features, physiological characteristics, interactions with the host, and variability in different survival conditions, it was difficult to comprehensively classify DPC species [33,34], and it was meaningless to distinguish DPC species only according to their host range [35]. Morphological differentiation was not enough to identify DPC at the species level. In order to clarify existing conflicts and increase the efficiency of *Diaporthe*/*Phomopsis* spp. identification, molecular methods were qualified for rapid and accurate identification of DPC species, particularly multi-locus phylogenetic analysis [7,9,21,29]. The related loci for identification of DPC species were mainly composed of the nuclear ribosomal internal transcribed spacer (ITS), partial translation elongation factor-1α (*tef1*), β-tubulin (*tub2*), *calmodulin* (*cal*), and histone H3 (*his3*) gene regions, and any combination of them can be used to analyze and identify DPC species [9,29].

In China, *Diaporthe*/*Phomopsis* spp. has been reported to cause three diseases in soybean, including stem blight, stem canker, and seed decay [24,25]. During a field survey in Qiubei County, Yunnan Province, China, wilt symptoms on chickpea crops were observed. The objective of the present study was to identify the pathogen species inciting the chickpea disease observed by using morphology and molecular analysis.

## 2. Results

### 2.1. Disease Symptoms

During the field surveys conducted in Qiubei County, Yunnan Province, we found the incidence of a new disease in chickpea plants. Of the diseased plants with obvious symptoms, about 20% were distributed in a few spots in the infected field (Figure 1A). The aerial parts of affected plants exhibited leaf chlorosis and wilting (Figure 1A). When affected plants were removed from the soil and dissected, they exhibited typical symptoms of basal stem rot, the exodermis of the basal stems and roots being brown to black and rotted, but there was no discoloration of the pith or vascular tissues (Figure 1B).

### 2.2. Pathogen and Morphological Characteristics

Three fungal isolates (ZD36-1, ZD36-2, and ZD36-3) were obtained from the diseased tissue. The isolates produced white colonies with lanose mycelia that occasionally turned light tan with age on PDA (Figure 2A,B). After three weeks, black stromata with pycnidia were produced. Pycnidia released yellow cirrus extruding masses of α-conidia (Figure 2C). The α-conidia (5.8–7.9 × 2.1–3.8 µm) were hyaline, nonseptate, ellipsoidal, and biguttulate in each cell (Figure 2D). No perithecia of the teleomorphs were formed on the PDA. To observe the teleomorphs of the pathogen, the diseased tissue fragments were subjected to incubation in a moist petri dish for 7 days. Masses of beaked perithecia were produced and embedded in the epidermis of the diseased tissue (Figure 2E). Perithecia (230–390 × 175–380 µm) were black, subglobose to globose, and had long necks (210 to 360 µm in length) (Figure 2E). Eight-spored asci (31.0 to 42.5 × 4.6 to 9.1 µm) were elongate-clavate and unitunicate-walled, with a refractive ring at the apex (Figure 2F). Ascospores (8.1 to 13.5 × 2.1 to 4.3 µm) were hyaline, smooth, fusoid or elongate-ellipsoid, two-celled, widest at the septum, tapering towards both ends, medianly septate, not constricted at the septum, with 1–2 guttules per cell (Figure 2F). These morphological features are consistent with those of *Diaporthe aspalathi* (Syn. *Diaporthe phaseolorum* var. *meridionalis*) [27].

### 2.3. Molecular Characteristics and Phylogenetic Analysis

The ITS, *tef1*, *tub2*, *cal*, and *his3* gene regions of rDNA from three isolates were sequenced using their respective common primers. Sequences of the ITS region and *EF1*-*α* gene obtained from the three isolates showed 99–100% similarity with more than 40 or several reported *D. aspalathi* strains in GenBank by a BLAST analysis. The aligned sequences of the five loci (ITS, *tef1*, *tub2*, *cal*, and *his3*) were cascaded, and 2655 bp sequences of the three isolates and other sequences from reported *Diaporthe*/*Phomopsis* spp. strains were involved in the dataset. 

The phylogenetic tree was constructed using a concatenated sequence dataset of the five gene regions (Figure 3). In the phylogenetic tree, the three isolates were grouped with those of three reference strains of *D. aspalathi*, whose host was rooibos (*Aspalathus linearis*) from South Africa in GenBank (Figure 3). The phylogenetic analysis revealed that the three isolates could be distinguished from other species of the *Diaporthe*/*Phomopsis* spp. Altogether, the results of the molecular characteristics and phylogenetic analysis strongly support that the three isolates clearly belong to *D. aspalathi* [36].

### 2.4. Pathogenicity and Host Range Tests

Firstly, the pathogenicity of the three *D. aspalathi* isolates was tested on the original host. The results showed that all isolates were strongly pathogenic to the original host, chickpea. All inoculated plants showed typical symptoms similar to those observed in the field, including leaf chlorosis, wilting, and basal stem rot. The diseased plants were dwarfed and wilted, eventually resulting in plant death (Figure 4A). There were no disease symptoms in the control plants. The pathogen was re-isolated from inoculated plants, thus fulfilling Koch’s postulates.

Secondly, the host range of the *D. aspalathi* isolate ZD36-3 was tested on eight other legume crops. The isolate ZD36-3 showed strong virulence in soybean, cowpea, and mung bean, and the symptoms were similar to those on chickpea, eventually causing plant death (Figure 4B–D). The ZD36-3 caused moderate symptoms on adzuki bean and common bean, with slightly enlarged spots at the inoculation site and with yellowing and wilting of leaves (Figure 4E–G). The ZD36-3 showed no pathogenicity in the remaining three crops, pea, faba bean, and grass pea, and no symptoms were observed on these three legume crops (Figure 4H–J). Thus, these results indicated that *D. aspalathi* from chickpea may cause the related disease in other legume crops, soybean, cowpea, mung bean, adzuki bean, and common bean.

## 3. Discussion

Chickpea is a nutrient-dense food that provides an abundance of protein, dietary fiber, and certain dietary minerals as animal feed and human food worldwide [37,38,39]. The biotic stresses in chickpea, particularly pathogens, are the primary cause of yield loss, and some of them, such as *Ascochyta rabiei* [40,41] and *Fusarium oxysporum* f.sp. *ciceris* [42,43] can cause up to 90%. In recent years, new diseases caused by pathogens have emerged continuously in legume crops. 

In this study, we discovered an emerging disease, basal stem rot, in chickpea from Qiubei County of Yunnan Province. It was found that basal stem rot of chickpea has been becoming one of the main diseases of chickpea in some regions. 

*Diaporthe* represents a highly complex genus containing numerous cryptic species. Many *Diaporthe* species that are morphologically similar proved to be genetically distinct, and several *Diaporthe* isolates that were previously identified based on their host were shown to represent different taxa. The genera *Diaporthe* and *Phomopsis* have received much taxonomic attention. Molecular identification has gradually replaced morphological identification as the main identification method for DPC species [9,21,29] because molecular identification is accurate and efficient and can avoid many controversial parts of morphological identification. However, morphological identification is still an important auxiliary means for DPC species identification. Brumer et al. [27] isolated *Diaporthe* and *Phomopsis* strains from soybean stems and seeds across South America from 1989 to 2014. Genomic DNA from 26 isolates underwent PCR-RFLP, AFLP analysis, and ITS sequencing. The molecular analysis identified *Phomopsis longicolla*, *D. phaseolorum* var. *sojae*, *D. caulivora*, and *D. aspalathi*. Pathogenicity tests of 13 *D. aspalathi* isolates revealed at least three races in Brazil. Both molecular and phenotypic analyses showed clustering based on collection date and pathogenicity, indicating pathogen variability.

In this study, the disease plant tissues with typical symptoms were collected, and three isolates with *Diaporthe*/*Phomopsis*-like fungi were isolated. Morphological identification indicated that the colony, conidia, perithecia, asci, and ascospore morphology were the same as those of *D. aspalathi*. Molecular identification showed that the ITS, *tef1*, *tub2*, *cal*, and *his3* gene regions of the three isolates were highly similar to previous *D. aspalathi* strains available in Genbank. The multi-locus phylogenetic analysis based on combining the five genes classified the three isolates into the *D. aspalathi* phylogenetic group (Figure 3). Therefore, based on the morphological characteristics, molecular characteristics, and multi-locus phylogenetic analysis, the three isolates ZD36-1, ZD36-2, and ZD36-3 were identified as *D. aspalathi*.

*Diaporthe* exhibits a wide geographic distribution and comprises pathogens, endophytes, and saprobes across diverse hosts [44,45]. *Diaporthe* species have been associated with economically significant hosts, such as *Citrus* sp. and *Coffea* sp., as well as ornamental plants, such as Clematis [44,46]. Some species have been reported as pathogens of die-backs, cankers, leaf spots, blights, melanoses, stem-end rot, and gummosis of various hosts [47,48,49,50,51]. For example, *D. kochmanii* and *D. kongii* have been associated with sunflower stem blight [7], while *D. australafricana* and *D. viticola* have been linked to cane and leaf spot disease of grapevine [52]. The *D. aspalathi*, formerly referred to as *D. phaseolorum* or *D. phaseolorum* var. *meridionalis* [53], was one of the most important DPC species and famous for causing southern stem canker on any growth stages of soybean with up to 100% yield loss worldwide, especially in South America [54], North America [28,53,55], and Africa [56,57]. Moreover, this pathogen, *D. aspalathi*, also induced die-back of rooibos (*Aspalathus linearis*), which caused substantial losses in South Africa [36]. Recently, *D. aspalathi* also induced die-back on honeybush (*Cyclopia longifolia*) [58].

In this study, nine legume crops were used in pathogenicity and host range tests. The three isolates were found to be strongly pathogenic to the original host chickpea. The representative isolate ZD36-3 showed highly pathogenic to other legume crops such as soybean, cowpea, and mung bean. It showed moderate or mild pathogenicity in adzuki bean and common bean, while no pathogenicity was observed in pea, faba bean, and grass pea. These results confirmed that *D. aspalathi* was not only a pathogen of soybean but also of chickpea, cowpea, mung bean, adzuki bean, and common bean.

Mostly, basal stem rot of plants is a soil-borne disease caused by soil-borne pathogens. But the pathogen *D. aspalathi* is also reported to be seed-borne, causing seed decay disease in soybean. Thus, the most possible reason for its severity is the extensive and transfer seed management [59]. The pathogen *D. aspalathi* causing basal stem rot on chickpea might come from preceding or fore-rotating crops, especially soybean, which is a main host of the pathogen.

Stanoeva Y and Beleva M [60] performed an investigation on the occurrence of *Phomopsis*/*Diaporthe* spp. on chickpea (*Cicer arietinum*) in Bulgaria. Their investigation is the first record of *C. arietinum* as a host of *Phomopsis*/*Diaporthe* spp. in Bulgaria. Unfortunately, *Phomopsis*/*Diaporthe* spp. was not identified at the species level only based on the cultural and morphological characteristics. Then, Thompson et al. [7] reported that *Diaporthe novem* of *Phomopsis*/*Diaporthe* complex was found to be associated with chickpea (*Cicer arietinum*) in Australia. In this study, we performed a detailed identification of the causal agent of basal stem rot from chickpea by morphology, molecular characteristics, multi-gene phylogenetic analysis, pathogenicity, and host range tests. Our results were sufficient to demonstrate that the three isolates from chickpea belong to *D. aspalathi*. This is the first report that *D. aspalathi* causes basal stem rot on chickpea in China and worldwide. Our study provided a foundation for disease control and screening of resistant germplasms in breeding programs of chickpea. And it also highlights that seed transport needs reinforcing quarantine and the disease of rotation legume crops needs reinforcing control.

## 4. Materials and Methods

### 4.1. Disease Survey

In late March 2017, a field disease survey of food legume crops was conducted by our research team in Yunnan Province, China. The survey discovered the wilt symptoms similar to those caused by *Diaporthe*/*Phomopsis* spp. of soybean plants in a chickpea field located in Qiubei County (23°63′ N, 103°34′ E). The incidence of the diseased plants with symptoms was about 20% in the infected field. The diseased plants were distributed in a few spots in the field. To confirm the causal agents, the diseased plants with typical symptoms of basal stem rot, such as leaf chlorosis and wilting, and the exodermis of the basal stems and roots being brown to black and rotted, were collected from these fields and used to subsequent pathogen isolation and identification.

### 4.2. Isolation of Pathogen from Diseased Plants

Diseased basal stems were washed under tap water and cut into several 2–3 mm sections from the margins of disease areas (1/3) or healthy parts (2/3). After being surface sterilized in 2% NaClO for 2 min, the tissues were subsequently rinsed three times with sterile distilled water and dried on sterilized filter paper. Every three to four pieces were cultivated on a potato dextrose agar (PDA; AoBoXing Biotech, Beijing, China). Plates were incubated at 25 °C under a 12 h photoperiod for 2–3 days. Fungal isolates were purified by transferring single hyphal tips to a fresh medium. The plates were incubated at 25 °C under a 12 h photoperiod. All isolates obtained were stored in 30% glycerol at −80 °C for long-term preservation.

### 4.3. Morphology Characteristics

Three isolates (ZD36-1, ZD36-2, ZD36-3) were used as representatives for morphological analysis. Five-millimeter-diameter mycelial plugs were cut from the edge of active colonies of each isolate by a puncher. The plugs were transferred to the center of a 90-mm-diameter plate containing PDA. Maintained at 25 °C under a 12 h light/12 h dark cycle. Teleomorph formation was induced from diseased chickpea tissue fragments, which were incubated in a moist petri dish. The conidia and ascospores produced were observed and measured under a light microscope (Olympus 31X, Tokyo, Japan) with the UV-C confocal imaging system (UVTEC).

### 4.4. Molecular Characteristics and Phylogenetic Analysis

The ZD36-1, ZD36-2, and ZD36-3 isolates were sub-cultured on cellophane-covered PDA for 10 days at 25 °C. The Mycelia of each isolate were scraped from the PDA plates with a sterilized blade. Genomic DNA was extracted by fusing the Fungi Genomic DNA Extraction Kit (Solarbio, Beijing, China), according to the manufacturer’s instructions. Purified DNA was suspended in 1× TE buffer and stored at −20 °C for later use. PCR assays were performed using specific primer pairs targeting different genomic regions: ITS region (primers ITS4/ITS5) [61], *tef1* gene (primers EF1-728F/EF1-986R) [62], *tub2* gene (primers Bt2a/Bt2b) [63], *cal* gene (primers CAL-228F/CAL-737R) [62], and *his3* gene (primers CYLH3F/H3-1b) [64]. PCR reactions were carried out using a Gene Amp 9700 thermocycler (Applied Biosystems, Foster City, CA, USA) in 50 μL reaction mixtures containing 25 ng of DNA, 2 μL of each primer, 25 μL of 2×Taq PCR Mastermix (TIANGEN, Beijing, China), and 17 μL ddH_2_O. The cycling protocol was as follows: an initial denaturation for 5 min at 95 °C, followed by 35 cycles of denaturation at 94 °C for 45 s, 45 s at 55 °C–65 °C (depending on the primer-specific annealing temperature), and extension at 72 °C for 60 s, with a final extension of 72 °C for 10 min. All PCR products were analyzed using 1.5% agarose gel with the Gelgreen Nucleic Acid Gel Stain (Biotium, Fremont, CA, USA). Then, a cloning method was applied to generate a sequence from the obtained amplicons by the five primer pairs, respectively (Sangon Biotech, Shanghai, China). The resulting sequences for each isolate were aligned with Multalin http://multalin.toulouse.inra.fr/multalin/ (accessed on 25 June 2023). Phylogenetic analysis was performed based on the concatenated multiple gene regions of the nuclear sequence dataset (ITS-*tef1*-*tub2*-*cal*-*his3*) using the MEGA X with Neighbor-Joining method and Tamura-Nei distance model with 1000 bootstrap replicates [29]. And BLAST online tool in the GenBank database (National Center for Biotechnology Information; NCBI) to compare sequence similarities (Appendix A).

### 4.5. Pathogenicity and Host Range Tests

Two pathogenicity trials were completed using chickpea seedlings (*Cicer arietinum*, original host), soybean (*Glycine max*), mung bean (*Phaseolus radiatus*), grass pea (*Lathyrus sativus*), common bean (*Phaseolus vulgaris*), cowpea (*Vigna unguiculata*), pea (*Pisum sativum*), adzuki bean (*Vigna angularis*), and faba bean (*Vicia faba*). The nine tested crops were sown in 500 mL paper cups filled with a mix of equal volumes of vermiculite and peat. The planted cups were randomly distributed on a greenhouse bench and incubated at 25 °C under a 12 h photoperiod and watered regularly every 3 to 4 days. The 12 to 14-day-old seedlings were inoculated using the modified hypocotyl puncture technique [11]. After the crops reached the stage of fully expanded true leaves, a sterile dissecting knife was used to make a light incision at hypocotyls of seedlings (not exceeding one-third of the stem diameter in width and approximately 1 cm in length), and hypocotyls of seedlings were inoculated with mycelial plugs. The plants inoculated with a sterile medium were used as negative controls. The treated plants were incubated in a misting room at 25 °C under high humidity (>90%) for 48 h. Misting occurred for 30 min every 2 h by an automatically controlled centrifugal humidifier, which maintained leaf wetness without excessive runoff. Then, the plants were removed to a greenhouse maintained at 25 °C until the plants were rated for disease symptoms. Plants were monitored every day for 4 weeks to detect symptoms. Pathogen was re-isolated from inoculated plants to verify Koch’s postulates. All tests were conducted twice.

## Figures and Tables

**Figure 1 plants-13-01950-f001:**
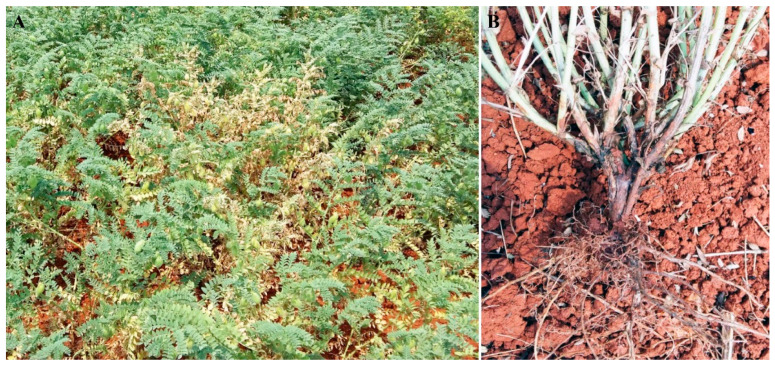
Symptoms of basal stem rot caused by *Diaporthe*/*Phomopsis* spp. on chickpea plants in the field. (**A**) Chickpea plants infected by *Diaporthe*/*Phomopsis* spp. showed leaf chlorosis and wilting; (**B**) exodermis of basal stems and roots become brown to black and rotted.

**Figure 2 plants-13-01950-f002:**
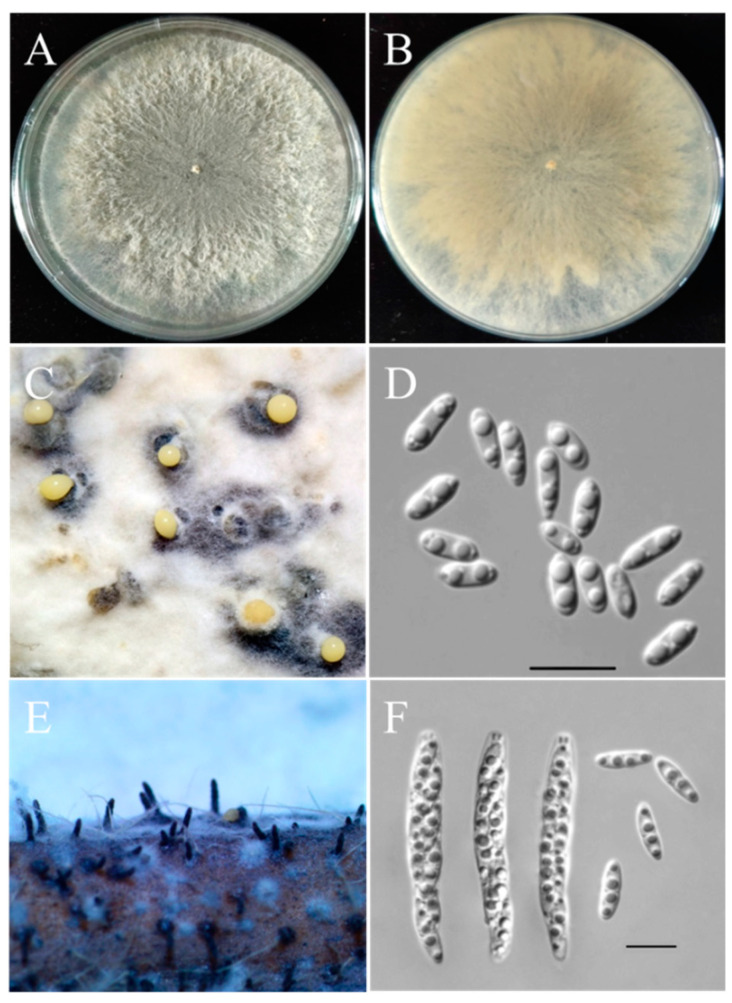
Morphological characteristics of the *Diaporthe aspalathi* isolates from chickpea. (**A**) Upper colony surface; (**B**) lower colony surface; (**C**) pycnidia extruding masses of conidia; (**D**) α-conidia with biguttulate; (**E**) embedded perithecia with visible beaks; (**F**) mature asci and ascospores with two-celled (Scales: 10 μm).

**Figure 3 plants-13-01950-f003:**
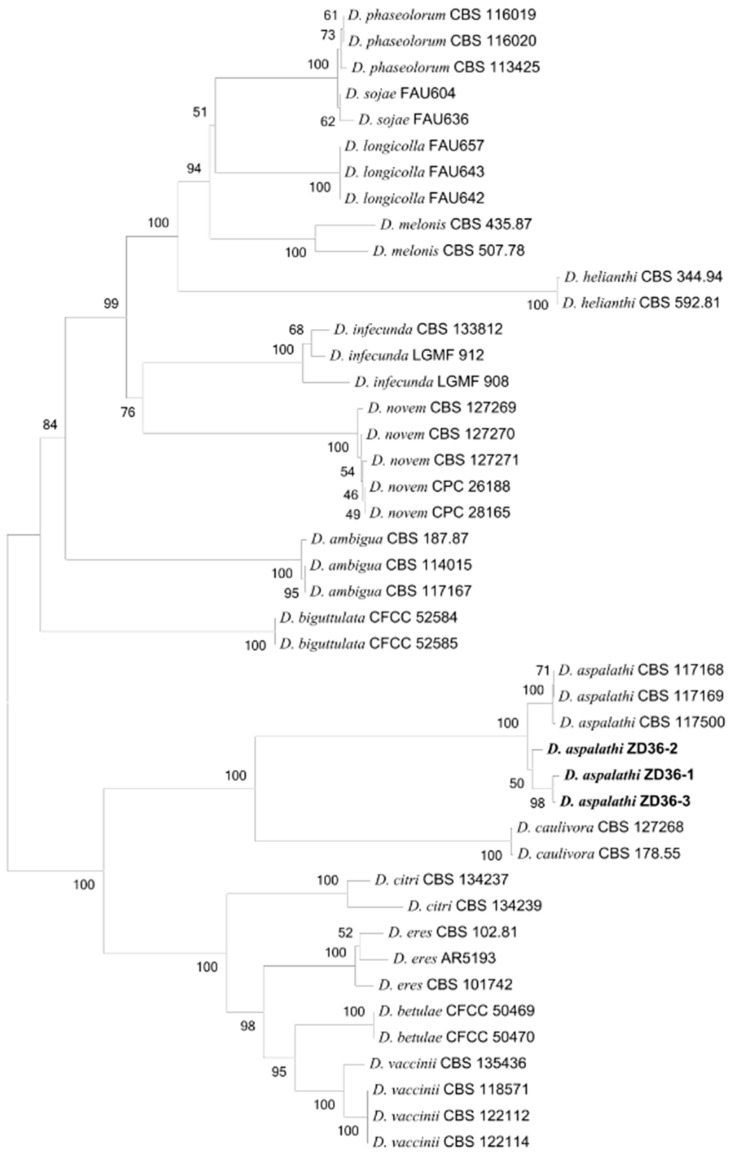
The Neighbor-Joining tree was generated from the analysis of the combined ITS, *tef1*, *tub2*, *cal*, and *his3* gene regions. The neighbor-joining bootstrapping values are shown above the branching nodes (>50% out of 1000 bootstraps). The strain number, host, and origin of each *Diaporthe* isolate were shown for each reference strain where available (The numbers mean neighbor-joining bootstrapping values).

**Figure 4 plants-13-01950-f004:**
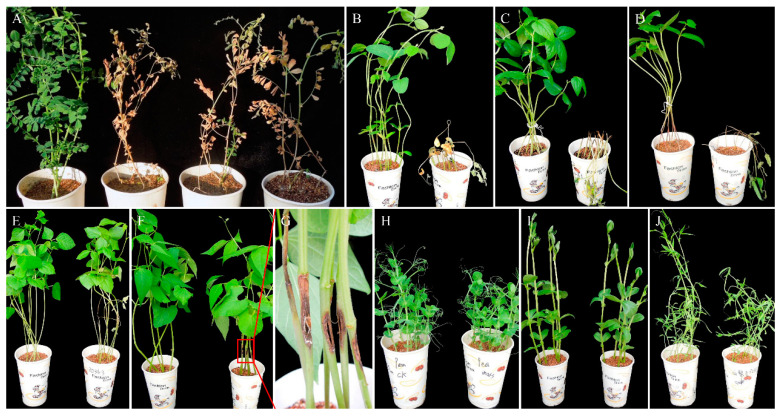
Pathogenicity and host range tests of *Diaporthe aspalathi* isolate causing chickpea stem canker. (**A**) Control and induced symptoms by inoculating the three isolates ZD36-1, ZD36-2, and ZD36-3 on chickpea seedlings. Control and induced symptoms on soybean (**B**), cowpea (**C**), mung bean (**D**), adzuki bean (**E**), common bean (**F**,**G**) (red box indicates inoculated stem showing extended lesion), pea (**H**), faba bean (**I**), and grass pea (**J**) by inoculating isolate ZD36-3.

## Data Availability

The data presented in this study are available on request from the corresponding author.

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
