# Peer review of "An Emerging Disease of Chickpea, Basal Stem Rot Caused by Diaporthe aspalathi in China"

_plants, 2024, doi:10.3390/plants13141950_

Round 1

Reviewer 1 Report

Comments and Suggestions for Authors

Dear authors, your record of a new disease associated to chickpea is interesting. However, unfortunately, to my opinion the manuscript cannot be accepted in the present form.  It needs of a deep review of the English language,  a better description of the material methods used and results. For example the section related to the symptoms assesment and re isolation after the pathogenicity test is not reported. I suggest to reviwe the manuscript and submit it again. I am sure you will get a manuscript more "complete and clear". Your efforts will be more effectively highlighted! 

Comments on the Quality of English Language

sentences are not clear, there are grammar mistakes and repetitions in the concepts

Author Response

Comments and Responses

Thanks for your detailed corrections and critical suggestions. We revised all points following your corrections and suggestions one by one, as follows.

  1. Line 13-14 As you suggested, we deleted “similar to those of wilt disease”, replaced “were” with “was” and added “in Qiubei County of Yunnan Province”.
  2. Line 15 As you suggested, we deleted “Thus, this study was performed to confirm the causal agent of the disease on chickpea”.
  3. Line 16-17 As you suggested, we replaced “The morphological characteristics of the isolates were observed and showed consistent with those of” with “Those isolates were morphologically found to be similar to”.
  4. Line18 As you suggested, we replaced “are” with “showed a” and deleted “those previous strains of”.
  5. Line22-23 As you suggested, we replaced “but no pathogenic to pea, faba bean, and grass pea. (Lathyrus sativus)” with “however, the isolates did not cause symtomps on grass pea (Lathyrus sativus).”, deleted “This study conducted a detailed identification of the causal agent for basal stem rot from chickpea by morphology, molecular characteristics, multi-gene phylogenetic analysis and pathogenicity host range tests. All results indicate that the isolates from chickpea belong to D. aspalathi.” and replaced “D.” with “Diaporthe”.
  6. Line31-33 As you suggested, we deleted “worldwide” and updated the data and rephrased the sentence, replaced “Worldwide, chickpea ranks third with accounting for 10.1 million tons annually behind beans (21.5 million tons) and peas (10.4 million tons) among the pulse crop from 2004 to 2013” with “Chickpea ranked third with an annual production of approximately 11.6 million tons following beans and peas in worldwide pulse production in 2020 (FAOSTAT, 2020)”.
  7. Line 290 We added space, thanks for your detail revision.

Reviewer 2 Report

Comments and Suggestions for Authors

Dear authors,

I had the pleasure to read the manuscript entitled "An Emerging Disease of Chickpea, Basal Stem Rot Caused by Diaporthe aspalathi in China".

The manuscript is well written, the topic is interesting and useful with several implications for the future.

I could suggest to change or modify slight things:

many times the authors report D. aspalath instead of D. aspalathi. Please corret

line 44: and Viroid (Pirovani et al 2014)

line 45: is the reference 4 enough updated?

Figure 1: is it possible to see healthy plants/roots beside infected plants?

Figure 3: captation should reports the meaning of the numbers in the tree.

lines 163-166: what about the roots you showed in the figure 1b? did you get the same isolates as before from the field? Please rephrase.

After few corrections and modification I recommend this manuscript to the journal.

Best regards

Author Response

  1. many times the authors report D. aspalath instead of D. aspalathi. Please correct.

Response: Thank you for noticing the error regarding "D. aspalath" in our manuscript. We have thoroughly reviewed the document and all D. aspalath were changed into "D. aspalathi".

  1. Line 44: and Viroid (Pirovano et al 2014)

Response: Thanks very much for your professional suggestion. We have now included the reference "and Viroid (Pirovano et al., 2014)".

  1. Line 45: is the reference 4 enough updated?

Response: Thanks very much for your careful and professional suggestions. We have updated the reference with the most recent literature in revised manuscript.

  1. Figure 1: is it possible to see healthy plants/roots beside infected plants?

Response: We can see the healthy plants around the infected plants in the same field. We are sorry for no taking photo of healthy plants. Unfortunately, it is not the chickpea growing season now, we are unable to obtain the picture. Thank you for your understanding.

  1. Figure 3: captation should reports the meaning of the numbers in the tree.

Response: Thanks very much for your careful and professional suggestions. We have revised the manuscript added the meaning of the numbers in the tree. Specifically, we have added the following statement:

"The Neighbor-Joining bootstrapping values are shown above the branching nodes (>50% out of 1000 bootstrap)."

  1. Lines 163-166: what about the roots you showed in the figure 1b? did you get the same isolates as before from the field? Please rephrase.

Response: Figure 1b showed the symptoms of infected plants. We didn’t get the same isolates from the field before. This is the first time obtain the isolates on chickpea.

Round 2

Reviewer 1 Report

Comments and Suggestions for Authors

authors improved the quality of the paper